# Effects of Inorganic Phosphorus-Solubilizing Bacteria on Rhizosphere Phosphorus Forms and Steroid Saponin Content of *Paris polyphylla* var. *yunnanensis*

**DOI:** 10.3390/biology14091284

**Published:** 2025-09-17

**Authors:** You Zhou, Yueheng Wang, Lingfeng Xu, Guoxin Lan, Dongqin Guo, Haizhu Zhang, Nong Zhou

**Affiliations:** 1Green Planting and Deep Processing of Genuine Medicinal Materials in Three Gorges Reservoir Area, Chongqing Engineering Laboratory, College of Biology and Food Engineering, Chongqing Three Gorges University, Chongqing 404120, China; zhyou0119@163.com (Y.Z.); jmswyh2023@126.com (Y.W.); 2Agricultural and Rural Committee of Kaizhou District, Chongqing 405499, China; xulingfeng953@126.com; 3College of Environmental and Chemical Engineering, Chongqing Three Gorges University, Chongqing 404120, China; feman7122@163.com; 4School of Pharmacy, Chongqing Three Gorges Medical College, Chongqing 404120, China; guodongqin1997@163.com; 5College of Pharmaceutical Science, Dali University, Dali 671003, China

**Keywords:** active ingredients, *Bacillus aryabhattai*, *Bacillus cereus*, medicinal plants, soil inorganic phosphorus

## Abstract

Three inorganic phosphorus-solubilizing bacteria strains were inoculated individually and in combination into the planting soil of the traditional Chinese medicinal herb *Paris polyphylla* var. *yunnanensis*, to find out the suitable strain or combination to improve the phosphorus utilization efficiency and medicinal quality of *Paris polyphylla* var. *yunnanensis*. The results showed that inorganic phosphorus-solubilizing bacteria inoculation increased the inorganic phosphorus content and proportion in the soil; the single-strain inoculation increased the effective phosphorus content in soil and the phosphorus content in the plant; there is an antagonistic effect when different strains are mixed for inoculation; the treatments of *B. aryabhattai* inoculation significantly increased the contents of total steroidal saponins in the plant. In summary, inorganic phosphorus-solubilizing bacteria inoculation improved the planting soil phosphorus form structure and medicinal quality of *Paris polyphylla* var. *yunnanensis*. These findings provide a basis for efficient phosphorus resource utilization and sustainable cultivation to enhance medicinal plant quality.

## 1. Introduction

Phosphorus is not only an indispensable nutrient for plant growth but also enhances the medicinal quality of numerous medicinal plants [1]. Plants primarily obtain phosphorus from the soil during their growth period. Soil phosphorus exists in two distinct forms: inorganic and organic. Inorganic phosphorus comprises about 60–80% of total phosphorus content in soils [2], making it the predominant phosphorus type. Traditionally, inorganic phosphorus has been categorized into aluminum phosphate (Al-P), iron phosphate (Fe-P), occluded phosphate (O-P), and calcium phosphate (Ca-P) [3,4]. Among these, Ca_2_-P, being water-soluble, is the most bioavailable phosphorus, making it easily assimilated by plants. Ca_8_-P, Al-P, and Fe-P are considered slow-release phosphorus sources that modulate soil phosphorus availability. In contrast, O-P and Ca_10_-P exist as insoluble compounds and serve as potential phosphorus reserves, making it difficult for plants to absorb directly. Applying phosphorus fertilizers can effectively address phosphorus deficiencies in soils. However, excessive phosphorus fertilizer application causes nutrient loss, disrupts soil nutrient balance, induces soil degradation and crusting, and damages the ecological environment [5,6]. Through mineral decomposition, inorganic phosphorus and other insoluble elements can be transformed into soluble forms, which plants can then absorb, thus promoting plant growth [7].

Phosphorus-solubilizing microorganisms (PSMs) represent a naturally abundant class of rhizosphere microorganisms beneficial to plant growth, significantly influencing the soil phosphorus cycle [8]. By releasing organic ions or protons, PSMs convert insoluble phosphorus into accessible forms, creating phosphorus-rich microenvironments within the rhizosphere and thereby improving phosphorus utilization efficiency in plants [9]. Among these microorganisms, phosphorus-solubilizing bacteria account for about half of the population, predominantly including the genera *Pseudomonas* and *Bacillus* [10].

*Paris polyphylla* var. *yunnanensis*, a perennial medicinal herb from the family *Liliaceae* (now classified as *Melanthiaceae*), mainly thrives in the Dali and Lijiang regions of Yunnan Province, China. The rhizome of *Paris polyphylla* var. *yunnanensis* has been officially recognized in the Pharmacopoeia of the People’s Republic of China due to its significant medicinal properties [11]. Characterized by bitter and cooling properties, this traditional Chinese medicine is utilized for heat-clearing, detoxification, hemostasis, cooling the liver, and immune regulation. Clinically, it has been extensively applied in treating conditions such as sore throat, bruises, abscesses, carbuncles, epidemic encephalitis B, lymph node tuberculosis, tonsillitis, and appendicitis. The herb contains diverse bioactive substances, notably polysaccharides and steroidal saponins [12,13,14]. According to the Pharmacopoeia standards, the rhizome’s quality is evaluated based on the content of four active saponins: polyphyllin I, polyphyllin II, polyphyllin VII, and dioscin [11]. Saponins isolated from *Paris polyphylla* var. *yunnanensis* exhibit potent anticancer effects, regulate protein and gene expression in humans, and possess antibacterial and hemostatic properties. Moreover, for the plant itself, saponins confer antioxidant, insecticidal, antifungal, antiparasitic, and antibacterial activities, enhancing plant resilience [15]. Previous studies have shown that inoculation with mycorrhizal fungi notably increases the total saponin content in *Paris polyphylla* var. *yunnanensis*, thereby boosting its medicinal quality [16]. Furthermore, the medicinal properties of this plant closely correlate with environmental factors [17], and reduced soil pH or organic matter can negatively impact its medicinal value [18]. Application of phosphorus-solubilizing bacteria significantly elevates phosphorus levels in plant tissues and enhances leaf protective enzyme activity [19]. The above results indicate that the application of inorganic phosphorus-solubilizing bacteria could enhance the medicinal quality of *Paris polyphylla* var. *yunnanensis* and merits further investigation.

In this study, three inorganic phosphorus-solubilizing bacterial strains were inoculated individually and in combination into the rhizosphere soil of *Paris polyphylla* var. *yunnanensis*. Measurements included phosphorus concentrations in the rhizosphere soil and plants. Relationships between soil and plant phosphorus contents were analyzed to evaluate whether phosphorus-solubilizing bacteria positively influence soil phosphorus composition. Additionally, steroidal saponin contents under various treatments were measured to determine if inoculation with inorganic phosphorus-solubilizing bacteria enhances the medicinal quality of *Paris polyphylla* var. *yunnanensis*, aiming to identify the most effective bacterial combination for its cultivation.

## 2. Materials and Methods

### 2.1. Objects and Experimental Design

The pot experiment was conducted in a greenhouse at Chongqing Three Gorges University, Chongqing, China. *Paris polyphylla* var. *yunnanensis* was planted from December 2023 to November 2024. Pots used for planting were 30 cm high, 28 cm top inner diameter, and 22 cm bottom inner diameter, each with a bottom tray. Each pot contained 5 kg of soil. Four-year-old dormant rhizomes with high uniformity and stable quality were obtained from a cultivation base in Baoshan, Yunnan, China (25°4′ N, 99°10′ E), ensuring consistent germplasm resources. The planting medium consisted of organic fertilizer, garden soil, and river sand mixed in a 1:1:2 ratio (pH 6.72, organic matter 21.2 mg·g^−1^, tototal N 1.83 mg·g^−1^, available N 10.64 mg·kg^−1^, total P 315 mg·g^−1^, available P 104.66 mg·kg^−1^, total K 8.83 mg·g^−1^, available K 28.64 mg·kg^−1^). Before planting, the soil was sieved through an 8 mm mesh, sterilized at 121 °C for 120 min, and subsequently incubated for one week. This study utilized three prominent inorganic phosphorus-solubilizing bacteria strains isolated from the rhizosphere of wild *Paris polyphylla* var. *yunnanensis* plants in Baoshan, Yunnan: *Bacillus cereus* strain Y1-1, and two *Bacillus aryabhattai* strains, Z6-1 and Z3-4.

In this experiment, seven treatment groups (S1–S7) and one control group (CK) were established, with ten pots per group and five seedlings per pot. Pre-cultured bacterial strains were prepared in physiological saline solution at a concentration of 106 CFU·mL^−1^. Each treatment pot received 300 mL bacterial suspension with different combinations of strains, while the CK group received 300 mL sterile physiological saline solution. The specific inoculation details are presented in Table 1.

### 2.2. Sampling

In November 2024, at the late stage of the annual growth cycle, three pots from each group were randomly selected for phosphorus and saponin determination. Rhizosphere soil from each of the five plants in a pot was collected by the shaking-root method, from 6 to 10 cm below the surface [20]. Rhizosphere soil samples from five plants per pot were pooled, air-dried, and sieved through an 80-mesh screen. The rhizomes collected from each pot were oven-dried at 35 °C (HGZF-II-101-0, Shanghai Yuejin Medical Equipment Co., Ltd., Shanghai, China) until reaching a stable weight. Afterward, they were combined, ground, and passed through an 80-mesh sieve. Each pot’s resulting rhizosphere soil and rhizome powders served as three biological replicates for the determination of phosphorus and saponin contents.

### 2.3. Determination of Steroid Saponins by UPLC

#### 2.3.1. Preparation of Standard Reference Solutions

Standard reference materials of polyphyllin I (CAS: 50773-41-6), polyphyllin II (CAS: 76296-72-5), polyphyllin VII (CAS: 68124-04-9), polyphyllin H (CAS: 81917-50-2), dioscin (CAS: 19057-60-4), and pseudoprotodioscin (CAS: 102115-79-7) were obtained from Chengdu Pusi Biotechnology Co., Ltd. (Chengdu, China, 2023). These compounds were individually dissolved in methanol (CAS: 67-56-1, Merck & Co., Inc., Shanghai, China, 2023) to prepare standard stock solutions with concentrations of 4.165, 3.245, 3.870, 2.975, 4.830, and 3.045 mg·mL^−1^, respectively, and stored at 4 °C [17].

#### 2.3.2. Preparation of Sample Solutions

*Paris polyphylla* var. *yunnanensis* rhizome powder samples (0.5 g) were extracted with 10 mL methanol in a conical flask via ultrasonication (SB-5200DTN ultrasonic cleaner, Ningbo Xinzhi Biotechnology Co., Ltd., Ningbo, China) for 30 min at room temperature, and the volume was adjusted to the required level with additional methanol. Subsequently, the extracts were centrifuged at 4000 r·min^−1^ for 15 min using a TDZ5-WS automatic balancing centrifuge (Hunan Saite Xiangyi Centrifuge Instrument Co., Ltd., Changsha, China). The resulting supernatants were filtered using a 0.22-µm microporous membrane (Guangdong huankai Microbial Technology Co., Ltd., Guangzhou, China), discarding the precipitates [17].

#### 2.3.3. Chromatographic Conditions

Steroidal saponin concentrations in rhizome samples were measured using an H-Class ultra-performance liquid chromatography (UPLC) system (Waters Inc., Milford, MA, USA) equipped with an Accucore PFP chromatographic column (Thermo Fisher Scientific Inc., Shanghai, China). Acetonitrile (CAS: 75-05-8, Merck & Co., Inc., Shanghai, China, 2023) (mobile phase A) and water (mobile phase B) constituted the mobile phase. The gradient elution was as follows: 0–5 min, 20–45% A; 5–9 min, 45–55% A; 9–18 min, 55–20% A; and 18–20 min, 20% A. A methanol-water solution was employed as the cleaning solvent. Chromatographic analysis conditions included a column temperature of 30 °C, a flow rate of 0.2 mL·min^−1^, a sample injection volume of 5 µL, and detection at a wavelength of 203 nm [17].

#### 2.3.4. Sample Determination

Rhizome powders from different treatment groups were prepared as described above, and steroidal saponin contents were determined according to the chromatographic conditions (Appendix A). The chromatographic profile is shown in Figure 1.

Six samples from the S6 group were tested using the above method. Peak areas were recorded, and the relative standard deviation (RSD, *n* = 6) of steroidal saponin peak areas ranged from 0.87% to 1.62%, indicating good repeatability.

The S6 sample solution was analyzed after storage periods of 0, 2, 4, 8, 12, and 24 h. The recorded peak areas showed RSD (*n* = 6) ranging from 0.77% to 2.57%, indicating sample solution stability for 24 h.

Known amounts of standard compounds were added to six S6 samples (0.25 g each) and analyzed according to the method described. Steroidal saponin recovery ranged from 96.8% to 102.35%, and RSD (*n* = 6) ranged from 1.19% to 2.52%, indicating good accuracy of the method.

### 2.4. Phosphorus Determination

Total phosphorus was quantified using the H_2_SO_4_-HClO_4_ digestion technique as previously reported [4]. The inorganic phosphorus levels were assessed following the method detailed by Koistinen [2].

### 2.5. Data Analysis

Statistical analyses were carried out using Excel 2010 for one-way ANOVA with a significance level set at *p* < 0.05. Correlation analyses and principal component analyses (PCA) were conducted using SPSS 22.0. Graphs and figures were created using Origin 2020 software.

## 3. Results

### 3.1. Effect of Inorganic Phosphorus-Solubilizing Bacteria on Phosphorus Fractions in the Rhizosphere Soil of Paris polyphylla var. yunnanensis

Figure 2 illustrates that single-strain inoculations (S1, S2, S3) significantly elevated Ca_2_-P levels (*p* < 0.05) in rhizosphere soil compared to multi-strain inoculations (S4, S5, S6, S7). A comparable trend was detected for Ca_8_-P concentrations. The highest Al-P levels were recorded in treatments inoculated with dual *Bacillus aryabhattai* strains (S6, S7). The Fe-P content exhibited no significant variation (*p* > 0.05) across most treatment conditions compared with the control (CK). Treatments S5, S6, and S7 exhibited significantly higher Ca_10_-P levels compared to CK, whereas Ca_10_-P in S2 showed no significant difference (*p* > 0.05) from CK. In contrast, S1, S3, and S4 had significantly lower Ca_10_-P concentrations (*p* < 0.05). The O-P content was higher (*p* < 0.05) in treatments S3, S4, and S6 compared to CK, while S1 presented a lower O-P level than CK. No significant differences (*p* > 0.05) in O-P were found among treatments S2, S5, S7, and CK.

Table 2 indicates that, apart from S4, the inorganic phosphorus contents in rhizosphere soils of all treatments were significantly greater (*p* < 0.05) than CK. Additionally, the proportion of inorganic phosphorus relative to total phosphorus was consistently higher in treatment groups compared to CK. Total phosphorus contents in the rhizosphere soils of treatments S1, S2, S3, and S6 were increased (*p* < 0.05) relative to CK. Notably, plant total phosphorus levels were significantly greater in single-strain inoculation treatments (S1, S2) compared with CK. Interestingly, dual-strain inoculation groups (S4 and S5) demonstrated relatively lower levels of soil inorganic phosphorus, total soil phosphorus, and plant total phosphorus compared to other groups.

### 3.2. Effect of Inorganic Phosphorus-Solubilizing Bacteria on Steroidal Saponin Content in Paris polyphylla var. yunnanensis

Figure 3 demonstrates that polyphyllin VII and total saponin contents in plants from most treatment groups, except S4 and S5, were increased (*p* < 0.05). Treatments S4 and S5 displayed relatively lower total saponin contents, aligning with the observed trends in soil and plant phosphorus content. Polyphyllin I, polyphyllin H, and dioscin concentrations were enhanced (*p* < 0.05) in most treatments. Conversely, pseudoprotodioscin concentrations across all treatment groups were lower (*p* < 0.05). Polyphyllin II contents were notably higher in treatments S3 and S6 (*p* < 0.05), while other treatments exhibited no significant variations (*p* > 0.05).

### 3.3. Principal Component Analysis (PCA) of Six Steroidal Saponins in Paris polyphylla var. yunnanensis Plants

PCA of steroidal saponin contents in plants from seven treatment groups and CK was conducted using SPSS software. As shown in Table 3, three principal components (PCs) with eigenvalues > 1.000 were extracted. PC1 had an eigenvalue of 1.791 and explained 29.848% of the variance, primarily representing polyphyllin I and polyphyllin VII. PC2 had an eigenvalue of 1.649 and explained 27.487% of the variance, mainly representing polyphyllin II and polyphyllin H. PC3 had an eigenvalue of 1.363 and explained 22.724% of the variance, primarily representing dioscin and pseudoprotodioscin. The cumulative variance explained by the three PCs was 80.058%.

The principal component scores for steroidal saponins in the treatment groups and CK group were calculated and ranked (Table 4) using the following formulas:
*F*_1_ = 0.513*X*_1_ + 0.037*X*_2_ + 0.061*X*_3_ − 0.106*X*_4_ + 0.554*X*_5_ − 0.033*X*_6_
*F*_2_ = 0.184*X*_1_ + 0.074*X*_2_ + 0.510*X*_3_ + 0.523*X*_4_ − 0.228*X*_5_ − 0.152*X*_6_
*F*_3_ = −0.064*X*_1_ + 0.561*X*_2_ + 0.249*X*_3_ − 0.232*X*_4_ + 0.058*X*_5_ + 0.472*X*_6_
*F_comprehensive_* = 0.373*F*_1_ + 0.343*F*_2_ + 0.284*F*_3_

In the above formulas, *F*1, *F*2, and *F*3 are scores for PC1, PC2, and PC3, respectively. *X*1–*X*6 represent standardized data for the six steroidal saponins. The comprehensive score (Fcomprehensive) was calculated using the ratios of variance contributions of the three PCs to the cumulative variance contribution as weights. As shown in Table 4, the treatment groups with the highest scores for PC1, PC2, and PC3 were S7, S6, and S5, respectively. Group S6 had the highest comprehensive score. The PC1, PC2, and comprehensive scores of all treatment groups were higher than those of CK.

### 3.4. Correlation Analysis Between Saponin Contents in Plants and Phosphorus in Rhizosphere Soils

As presented in Table 5, the concentration of polyphyllin I in plants exhibited a highly significant positive correlation with the contents of Ca_10_-P and O-P in rhizosphere soils (*p* < 0.01). A significant positive correlation was also observed between polyphyllin II concentration and O-P content (*p* < 0.05). Polyphyllin VII concentrations showed highly significant positive correlations with Ca_10_-P and soil inorganic phosphorus (*p* < 0.01), and a significant positive correlation with total soil phosphorus (*p* < 0.05). Dioscin levels were significantly positively associated with Ca_2_-P and total soil phosphorus (*p* < 0.05), highly positively associated with Ca_8_-P (*p* < 0.01), and exhibited a highly significant negative correlation with Ca_10_-P content (*p* < 0.01). Pseudoprotodioscin concentrations displayed highly significant positive correlations with Al-P and total soil phosphorus (*p* < 0.01). Total plant saponin content had significant positive correlations with soil inorganic and total phosphorus contents (*p* < 0.05), and a highly significant positive correlation with Al-P content (*p* < 0.01). Generally, most plant saponin contents were positively correlated with rhizosphere soil phosphorus contents, although polyphyllin II showed a negative correlation with most phosphorus forms in rhizosphere soils. Plant phosphorus content demonstrated highly significant positive correlations with Ca_8_-P, inorganic soil phosphorus, total soil phosphorus, pseudoprotodioscin, and total plant saponins (*p* < 0.01). Additionally, it was positively correlated with polyphyllin VII and polyphyllin H contents (*p* < 0.05) and significantly negatively correlated with O-P content (*p* < 0.05).

## 4. Discussion

Previous studies have reported that the utilization rate of phosphorus absorbed by plants is only 15–25% in soils receiving long-term large-scale chemical fertilizer application [21]. The low phosphorus utilization rate leads to the accumulation of various phosphorus forms in soils. Certain *Bacillus* isolates can mineralize organic phosphorus, an ability associated with genes encoding phytase [22]. Moreover, *Bacillus* can secrete various organic acids, thus dissolving insoluble phosphorus [23]. In this study, the rhizosphere soils of all treatment groups showed significantly higher Ca_2_-P (water-soluble phosphorus and the most plant-available form) compared to CK. Ca_2_-P content in single-strain inoculation groups (S1, S2, and S3) was significantly higher than that in mixed-strain groups. Previous studies indicated that the order of phosphorus contents in soils is typically Ca_10_-P > Ca_8_-P > O-P > Al-P > Fe-P > Ca_2_-P [3,4], with Ca_10_-P (a potential phosphorus source difficult for plants to absorb directly) accounting for a large proportion. In contrast, this study showed that Ca_10_-P contents in S1, S2, and S3 were lower than Ca_8_-P contents (a slow-acting form convertible to available phosphorus). These results suggest that inoculation with the three *Bacillus* strains effectively improves soil phosphorus structure and increases available phosphorus content. Previous studies reported interspecific antagonism in co-cultures of different *Bacillus* strains [24]. When multiple *Bacillus* species establish molecular interactions with plants, their quorum-sensing signals can interfere with each other, weakening their respective interactions with the plants. Compared to single-strain cultures, mixed-strain inoculations showed significantly lower relative abundance, propagation rates, metabolite production, and beneficial effects on plant growth. Similarly, Wang [25] reported significantly lower alkane degradation rates by mixed *Bacillus* strains compared to single strains, likely due to resource competition. Therefore, the significantly higher contents of effective phosphorus (Ca_2_-P) and slow-acting phosphorus (Ca_8_-P) in single-strain groups (S1, S2, S3) compared to mixed groups observed in this study could also be explained by interspecific antagonism among different *Bacillus* strains.

In this study, soil inorganic phosphorus proportions in treatment groups were significantly higher. This result indicates that the bacterial strains effectively converted soil organic phosphorus into inorganic forms. Plant total phosphorus contents in single-strain groups (S1, S2) were significantly higher than CK. However, no significant differences or significantly lower contents occurred in the plant total phosphorus of mixed-strain groups (S3–S7) compared to CK. This result may also be due to interspecific antagonism among *Bacillus* strains. For groups S1 and S2, inoculated individually with *B. cereus* and *B. aryabhattai*, respectively, the plant total phosphorus content was significantly higher than in CK. *B. cereus* not only dissolves inorganic phosphorus but also organic phosphorus. It produces oxalic, malonic, and succinic acids, secretes acidic, neutral, and alkaline phosphatases, and produces IAA, thereby promoting plant growth [26]. Additionally, *B. cereus* can enhance plant integration with arbuscular mycorrhizal fungi (AMF), significantly increasing plant biomass when co-inoculated with AMF [27]. Interestingly, despite significantly lower total soil phosphorus contents in groups S3, S5, and S7 compared to CK, their inorganic phosphorus contents were significantly higher. Each of these groups contained at least one strain of *B. aryabhattai*, which functions as both a soil phosphorus-solubilizing bacterium and an endophyte [28]. *B. aryabhattai* can collaborate with AMF, promoting conversion of soil organic phosphorus into inorganic forms [29]. As an endophytic bacterium with plant growth-promoting properties, *B. aryabhattai* can solubilize inorganic and organic phosphorus, as well as potassium, and produce IAA, thereby enhancing plant growth and phosphorus absorption [30]. The highly significant positive correlation between soil inorganic phosphorus, plant phosphorus, and total saponin content observed in this study further indicates that *B. aryabhattai* plays an essential role in enhancing the medicinal quality of *Paris polyphylla* var. *yunnanensis*.

Total Paris saponins can inhibit cell proliferation in lung cancer, nasopharyngeal carcinoma, and gastric cancer [31]. Polyphyllin I inhibits lung and colon cancer cells [32]. Polyphyllin VI and polyphyllin VII inhibit liver cancer cell growth [33]. Polyphyllin I, polyphyllin II, polyphyllin VI, and polyphyllin VII significantly inactivate influenza virus type A by blocking viral adsorption, invasion, and proliferation in target cells [34]. Polyphyllin H exhibits anti-tumor, anti-inflammatory, and anticoagulant activities [13]. Through the A1 and A3 adenosine receptor pathways, it inhibits glioma U251 cell proliferation, exerts anti-inflammatory and analgesic effects [35], and increases serum SOD activity [36]. Dioscin has anti-inflammatory effects, inhibits multiple cancer cell types, and alleviates Alzheimer’s disease [37]. Pseudoprotodioscin inhibits endometrial cancer cell growth, repairs heart damage, and improves liver fat metabolism and antioxidant function in rats [38,39,40]. In this study, plants inoculated with *B. aryabhattai* (groups S2, S3, S6) had significantly higher total steroidal saponin contents and comprehensive principal component scores compared to other groups. *Bacillus* species are well-known as plant probiotics or pathogens; however, their primary agricultural role is as producers of biological control agents (fungicides, bactericides, fertilizers) [20]. Beyond phosphorus solubilization, previous studies have reported that inoculation with *B. aryabhattai* alters the microbial community structure within the plant rhizosphere. Such inoculation significantly elevates the abundance of beneficial rhizobacteria, fostering positive interactions between the rhizosphere microbiome and plants, consequently promoting plant growth and improving quality [28]. Furthermore, *B. cereus* produces the enzyme ACC (1-aminocyclopropane-1-carboxylic acid) deaminase, which breaks down the ethylene precursor ACC into α-ketobutyric acid and ammonia. This enzymatic activity lowers ethylene levels in plants under stress, enhancing plant resistance to adverse conditions [41]. However, ethylene directly affects saponin accumulation and expression of squalene synthase and squalene cyclooxygenase genes during saponin biosynthesis [42]. For instance, treating cultured ginseng cells (*Panax ginseng*) with ethylene precursor ACC induces transcriptional expression of squalene synthase and squalene cyclooxygenase, increasing saponin content [43]. Therefore, ACC deaminase produced by *B. cereus* in this study could explain the relatively low saponin contents observed in groups inoculated with *B. cereus* (S1, S4, S5).

## 5. Conclusions

In this study, three inorganic phosphorus-solubilizing bacterial strains (*Bacillus cereus* Y1-1, *Bacillus aryabhattai* Z6-1, and *Bacillus aryabhattai* Z3-4) were inoculated individually and in combination into the rhizosphere soil of *Paris polyphylla* var. *yunnanensis*. Results showed that inoculation with inorganic phosphorus-solubilizing bacteria improved the phosphorus structure in soil and increased the available phosphorus content. Single-strain inoculations (S1, S2, and S3) significantly enhanced the effective phosphorus (Ca_2_-P) content in rhizosphere soil. Antagonism among different strains was indicated by the lower Ca_2_-P and Ca_8_-P contents in mixed-strain groups compared to single-strain groups, reflecting competition for resources or production of inhibitory substances. Inoculation with *B. aryabhattai* (S2, S3, and S6) significantly increased total steroidal saponin contents in plants. The relatively lower saponin contents in groups S1, S4, and S5 may be due to ACC deaminase production by *B. cereus*. In summary, inoculating inorganic phosphorus-solubilizing bacteria markedly improved phosphorus bioavailability in the rhizosphere soil and enhanced the medicinal quality of *Paris polyphylla* var. *yunnanensis*. The results presented here support the efficient utilization of phosphorus resources and sustainable agricultural practices aimed at optimizing medicinal plant quality. Subsequent studies will further explore the impact of phosphorus-solubilizing bacterial inoculation on the rhizosphere microbiome, transcriptomic and metabolomic profiles, and the biosynthesis pathway of saponins, to elucidate the underlying mechanisms through which these microorganisms enhance yield and medicinal properties in *Paris polyphylla* var. *yunnanensis*.

## Figures and Tables

**Figure 1 biology-14-01284-f001:**
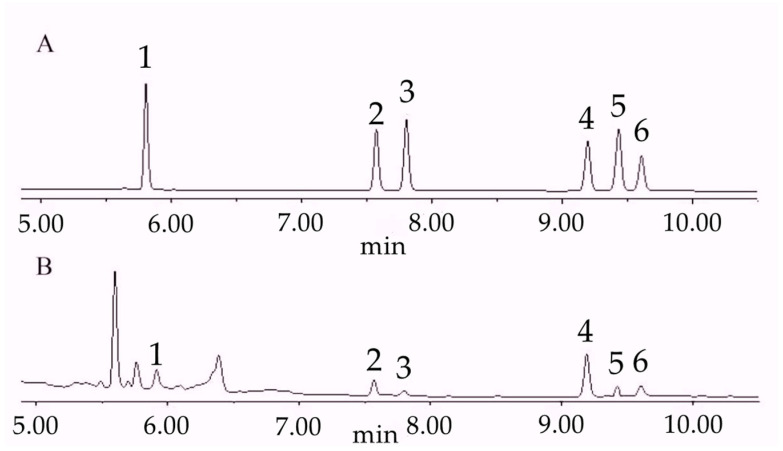
Chromatogram of steroid saponins. (**A**): chromatogram of standard reference; (**B**): chromatogram of sample (S6). 1: pseudoprotodioscin; 2: polyphyllin VII; 3: polyphyllin H; 4: polyphyllin II; 5: dioscin; 6: polyphyllin I.

**Figure 2 biology-14-01284-f002:**
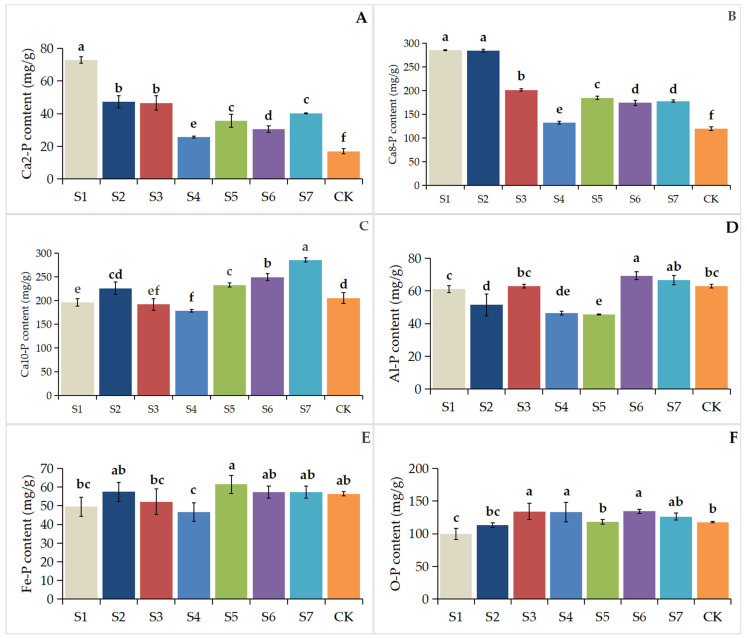
Contents of different inorganic phosphorus forms in the rhizosphere soil of *Paris polyphylla* var. *Yunnanensis*. (**A**): Ca_2_-P content; (**B**): Ca_8_-P content; (**C**): Ca_10_-P content; (**D**): Al-P content; (**E**): Fe-P content; (**F**): O-P content. S1: *B. cereus* Y1-1; S2: *B. aryabhattai* Z6-1; S3: *B. aryabhattai* Z3-4; S4: *B. cereus* Y1-1 + *B. aryabhattai* Z6-1; S5: *B. cereus* Y1-1 + *B. aryabhattai* Z3-4; S6: *B. aryabhattai* Z6-1 + *B. aryabhattai* Z3-4; S7: *B. cereus* Y1-1 + *B. aryabhattai* Z6-1 + *B. aryabhattai* Z3-4; CK: saline solution. Different letters indicate significant differences at *p* < 0.05 levels.

**Figure 3 biology-14-01284-f003:**
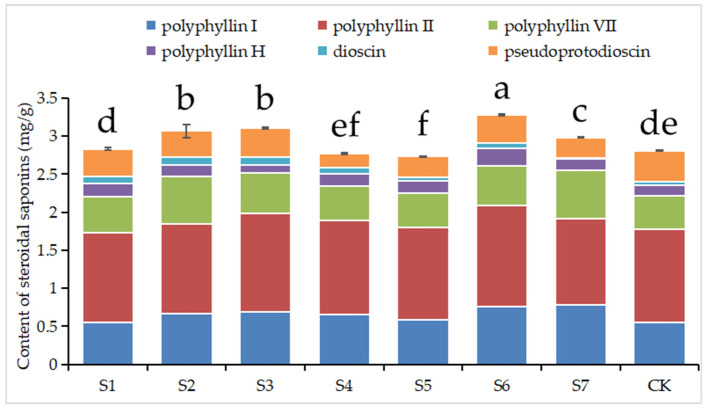
Contents of six steroidal saponins in *Paris polyphylla* var. *yunnanensis* plants under different treatments. Different letters indicate significant differences at *p* < 0.05 levels.

**Table 1 biology-14-01284-t001:** Treatment groups and their inoculated bacterial strains.

Group	Inoculated Strain
S1	*Bacillus cereus* Y1-1, 300 mL
S2	*Bacillus aryabhattai* Z6-1, 300 mL
S3	*Bacillus aryabhattai* Z3-4, 300 mL
S4	*Bacillus cereus* Y1-1, 150 mL + *Bacillus aryabhattai* Z6-1, 150 mL
S5	*Bacillus cereus* Y1-1, 150 mL + *Bacillus aryabhattai* Z3-4, 150 mL
S6	*Bacillus aryabhattai* Z6-1, 150 mL + *Bacillus aryabhattai* Z3-4, 150 mL
S7	*Bacillus cereus* Y1-1, 100 mL + *Bacillus aryabhattai* Z6-1, 100 mL + *Bacillus aryabhattai* Z3-4, 100 mL
CK	saline solution, 300 mL

**Table 2 biology-14-01284-t002:** Phosphorus contents (mg·g^−1^) in rhizosphere soils and plants of *Paris polyphylla* var. *yunnanensis*.

Group	Soil Inorganic Phosphorus	Soil Total Phosphorus	Plant Total Phosphorus	Soil Inorganic Phosphorus Proportion
S1	767.718 ± 22.493 ab	1061.850 ± 31.111 b	1261.850 ± 31.111 a	72.30%
S2	780.070 ± 21.394 a	1125.642 ± 30.872 a	1340.642 ± 30.872 a	69.30%
S3	689.514 ± 3.252 d	953.685 ± 4.498 d	1073.685 ± 4.498 b	72.30%
S4	562.889 ± 9.947 f	768.974 ± 13.588 g	1028.974 ± 13.588 d	72.20%
S5	680.814 ± 11.337 d	838.441 ± 13.962 f	1089.441 ± 13.962 c	81.20%
S6	715.742 ± 12.980 c	1000.617 ± 18.147 c	1231.617 ± 18.146 b	71.53%
S7	754.234 ± 7.359 b	916.445 ± 8.942 e	1152.445 ± 8.942 b	82.30%
CK	589.085 ± 12.162 e	933.574 ± 19.274 de	1171.574 ± 19.274 b	63.10%

Note: Different letters in the same column indicate significant differences at *p* < 0.05 levels.

**Table 3 biology-14-01284-t003:** Principal component analysis matrix of steroidal saponins.

		Principal Component 1	Principal Component 2	Principal Component 3
Steroidal saponin	polyphyllin VII	0.919	−0.283	0.006
	polyphyllin I	0.911	0.342	−0.132
	polyphyllin II	0.14	0.83	0.43
	polyphyllin H	−0.095	0.778	−0.286
	dioscin	0.014	0.181	0.861
	pseudoprotodioscin	−0.125	−0.19	0.708
Eigenvalue		1.791	1.649	1.363
Variance contribution rate (%)		29.848	27.487	22.724
Cumulative variance contribution rate (%)		29.848	57.335	80.058

**Table 4 biology-14-01284-t004:** Principal component scores and rankings of steroidal saponins in *Paris polyphylla* var. *yunnanensis* plants among different groups.

Group	*F* _1_	Sort	*F* _2_	Sort	*F* _3_	Sort	*F_comprehensive_*	Sort
S1	−1.044	7	−0.024	4	0.343	3	−0.300	6
S2	0.929	2	−0.777	6	0.794	2	0.306	3
S3	0.517	4	−0.118	5	1.673	1	0.627	2
S4	−0.380	5	0.743	2	−0.818	7	−0.119	4
S5	−0.901	6	0.186	3	−0.675	6	−0.464	7
S6	0.553	3	1.952	1	0.179	4	0.927	1
S7	1.479	1	−0.811	7	−1.515	8	−0.157	5
CK	−1.153	8	−1.152	8	0.020	5	−0.820	8

**Table 5 biology-14-01284-t005:** Correlation analysis between saponin contents in plants and phosphorus contents in rhizospheric soils.

	Ca_2_-P	Ca_8_-P	Ca_10_-P	Al-P	Fe-P	O-P	Soil Inorganic Phosphorus	Soil Total Phosphorus	Plant Phosphorus
Polyphyllin I	−0.144	−0.092	0.527 **	0.388	0.105	0.593 **	0.286	0.016	0.083
Polyphyllin II	−0.270	−0.314	−0.296	0.154	−0.096	0.480 *	−0.330	−0.142	−0.088
Polyphyllin VII	0.198	0.394	0.605 **	0.297	0.236	0.067	0.668 **	0.490 *	0.440 *
Polyphyllin H	0.017	0.024	0.218	0.145	0.015	0.005	0.135	0.071	0.406 *
Dioscin	0.420 *	0.562 **	−0.648 **	−0.179	−0.302	−0.058	0.178	0.462 *	0.234
Pseudoprotodioscin	0.177	0.266	−0.026	0.579 **	0.158	−0.247	0.264	0.680 **	0.527 **
Total saponin	0.062	0.210	0.309	0.574 **	0.124	0.365	0.447 *	0.510 *	0.505 **
Plant phosphorus	0.328	0.604 **	0.317	0.345	0.230	−0.405 *	0.653 **	0.866 **	\

Note: “*” indicates significant correlation (*p* < 0.05); “**” indicates extremely significant correlation (*p* < 0.01).

## Data Availability

The data presented in this study are available at reasonable request to the corresponding author.

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
