# Peer review of "Effects of Inorganic Phosphorus-Solubilizing Bacteria on Rhizosphere Phosphorus Forms and Steroid Saponin Content of Paris polyphylla var. yunnanensis"

_biology, 2025, doi:10.3390/biology14091284_

Round 1
Reviewer 1 Report
Comments and Suggestions for Authors
Journal: Biology (ISSN 2079-7737)
Manuscript ID: biology-3839318
Title: Effects of inorganic phosphorus solubilizing bacteria on the rhizosphere phosphorus form and steroid saponin content of Paris polyphylla var. yunnanensis
This is a well-referenced and well-written article. The manuscript findings may add to the existing knowledge. The practical significance of this manuscript, which links the improvement of phosphorus utilization efficiency with the quality of an important medicinal plant (Paris polyphylla var. yunnanensis), which makes it of practical value in sustainable agriculture and its medicinal use. Its experimental design, which utilized three different isolates of phosphorus-solubilizing bacteria (PSB), provides good treatment diversity and allows for useful comparisons between the effects of individual strains and their mixtures. The results also highlighted the positive effects of Bacillus aryabhattai strains on increasing steroidal saponin content, an important addition from a pharmacological perspective. However, the following can be taken into consideration to further improve the quality of the manuscript:
- The sentences must be clear, as some sentences are very long and need to be divided to facilitate reading, especially when describing the treatments (S1, S2, S3…). In the abstract, the methodological part also lacked mention of the number of replicates, growth conditions (experimental period, etc.), or methods for measuring phosphorus and saponins
- In addition, when presenting the results in the abstract, a “competitive effect” was mentioned when the strains were used together, but the mechanism was not explained or the possible causes discussed (e.g., competition for resources or secretion of inhibitory substances). This should also be addressed in more depth in the discussion. The authors did not provide numerical results or percentages for the most important results in the manuscript in the abstract section.
- The concluding sentence is very general and broad in its conclusion. It would be better to link the results more closely to the mechanism of bacterial action and its role in improving phosphorus utilization efficiency and plant quality.
- Plz choose expressive and important keywords, while not repeating any word mentioned in the title of the manuscript. Plz write the keywords in alphabetical order. Also, plz capitalize the first letter of each word.
- L37-39: “Plants require the participation of many elements in their growth process, in which phosphorus plays an important role not only in composing organic compounds of plant body, but also participating in metabolic processes [1]” Plz delete this sentence and other sentences that do not add to the manuscript because they are general and well-known information. Write about phosphorus and its importance directly in a way that serves the work in the current manuscript.
- L60: “phosphate-solubilizing microorganisms (PSMs)” Plz define all acronyms in the abstract (L23) when they appear first. If they appear once, there is no need to use acronyms.
- The Hypothesis should be obvious at the end of the introduction part.
- L126: “2.3 Experimental Design” The authors did not mention in this section the type of statistical design for the experiment, the number of replications, or other important points that should be detailed in this section.
- L195: “2.7 Data Analysis” Plz write the method used to test the averages of the results to determine whether the results are significant or not.
- L274: “Figure 2.” The design of this figure and the results of the higher grade columns make the shape not good; only the significant letters should be included, and this figure should be redesigned.
- Restrict the results part to only results (values etc.) and remove all sentences which actually belong to the discussion part.
- The results were separated without a storyline or connection between each other, plz revise and improve.
- Though the discussion section is written well, however authors still need to add cool logical reasoning supported with study findings.
- In the conclusion sections, authors must also add the future research directions that must be adapted to this aspect.
- The manuscript contains grammatical and syntax errors that can distract the readers; therefore, the quality of English must be improved in the updated version.
The English could be improved to more clearly express the research.
Author Response
- The sentences must be clear, as some sentences are very long and need to be divided to facilitate reading, especially when describing the treatments (S1, S2, S3…). In the abstract, the methodological part also lacked mention of the number of replicates, growth conditions (experimental period, etc.), or methods for measuring phosphorus and saponins.
Response: Thank you for pointing out this, we have made revision according to your advise (Line 35-52).
- In addition, when presenting the results in the abstract, a “competitive effect” was mentioned when the strains were used together, but the mechanism was not explained or the possible causes discussed (e.g., competition for resources or secretion of inhibitory substances). This should also be addressed in more depth in the discussion. The authors did not provide numerical results or percentages for the most important results in the manuscript in the abstract section.
Response: We think this is an excellent suggestion, we have added the possible mechanism of “competitive effect” in the section of discussion (Line 395-403), and added the percentage of incremental related data in the abstract (Line 43-46).
- The concluding sentence is very general and broad in its conclusion. It would be better to link the results more closely to the mechanism of bacterial action and its role in improving phosphorus utilization efficiency and plant quality.
Response: We agree with the comment and re-wrote the conclusion (Line 464-485).
- Plz choose expressive and important keywords, while not repeating any word mentioned in the title of the manuscript. Plz write the keywords in alphabetical order. Also, plz capitalize the first letter of each word.
Response: Thank you for your suggestion. As suggested by reviewer, we have re-chosen the key words in the revised manuscript (Line 53-54).
- L37-39: “Plants require the participation of many elements in their growth process, in which phosphorus plays an important role not only in composing organic compounds of plant body, but also participating in metabolic processes [1]” Plz delete this sentence and other sentences that do not add to the manuscript because they are general and well-known information. Write about phosphorus and its importance directly in a way that serves the work in the current manuscript.
Response: We deeply appreciate the reviewer’s suggestion. We re-wrote this part in the revised manuscript (Line 58-59).
- L60: “phosphate-solubilizing microorganisms (PSMs)” Plz define all acronyms in the abstract (L23) when they appear first. If they appear once, there is no need to use acronyms.
Response: We have made revisions according to the reviewer's comments.
- The Hypothesis should be obvious at the end of the introduction part.
Response: We are grateful for the suggestion and re-wrote this part according to the reviewer’s advise (Line 108-117).
- L126: “2.3 Experimental Design” The authors did not mention in this section the type of statistical design for the experiment, the number of replications, or other important points that should be detailed in this section.
Response: We are so sorry for our negligence, we have re-written the section of “2.2 Sampling”, and provided these information in that section: “at the late stage of the annual growth cycle, three pots from each group were randomly selected for phosphorus and saponin determination. Rhizosphere soil from each of the five plants in a pot was collected by the shaking-root method, from 6-10 cm below the surface. Rhizosphere soil from each of the five plants in a pot was collected by the shaking-root method, from 6-10 cm below the surface. Rhizosphere soils samples from five plants per pot were pooled, air-dried, and sieved through an 80-mesh screen. The rhizomes collected from each pot were oven-dried at 35℃ until reaching a stable weight. Afterward, they were combined, ground, and passed through an 80-mesh sieve. Each pot's resulting rhizosphere soil and rhizome powders served as three biological replicates for the determination of phosphorus and saponin contents.” (Line 146-155).
- L195: “2.7 Data Analysis” Plz write the method used to test the averages of the results to determine whether the results are significant or not.
Response: We are sorry for this mistake and we made revision according to your comment (Line 206-210).
- L274: “Figure 2.” The design of this figure and the results of the higher grade columns make the shape not good; only the significant letters should be included, and this figure should be redesigned.
Response: Thank you for your advise, we have redesigned this figure and only the significant letters were retained (Line 263-296, Figure 2).
- Restrict the results part to only results (values etc.) and remove all sentences which actually belong to the discussion part.
Response: We apologize for our negligence, we have replaced these sentence according to the reviewer’s suggestion.
- The results were separated without a storyline or connection between each other, plz revise and improve.
Response: We are grateful for the suggestion. We have made revision in the manuscript.
- Though the discussion section is written well, however authors still need to add cool logical reasoning supported with study findings.
Response: We deeply appreciate the reviewer’s suggestion. We have made revision in the manuscript.
- In the conclusion sections, authors must also add the future research directions that must be adapted to this aspect.
Response: Thank you for the suggestion. We have added the future research directions at the end of section conclusion according to the reviewer’s advise (Line 481-485).
- The manuscript contains grammatical and syntax errors that can distract the readers; therefore, the quality of English must be improved in the updated version.
Response: Thank you for your careful review. We are very sorry for the mistakes in this manuscript and inconvenience they caused in your reading. The manuscript has been thoroughly revised and edited by a professional company, so we hope it can meet the journal’s standard. In addition, we provided the editing certificate in the attachment.

Reviewer 2 Report
Comments and Suggestions for Authors
Dear authors! Thank you for the article. It is devoted to the study of the role of soil inoculation of the medicinal plant Paris polyphylla var. Yunnanensis with three strains of Bacillus cereus Y1-1, Bacillus aryabhattai Z6-1, Bacillus aryabhattai Z3-4 to enhance their medicinal effects by influencing the content of saponins.
There are a number of comments and recommendations, the consideration of which will contribute to the improvement of the article, I have listed them in the relevant sections of the review.
- Key words should not duplicate words included in the title of the article.
- The Introduction does not contain enough information about the role of the studied saponins in the life of the plant. I consider it necessary to add this information to the article.
- It is desirable to more clearly define the purpose of the study in the Introduction. In its current form, it is written rather vaguely in lines 94-102 and after reading this text it is difficult to understand what was most important in this study.
- Paragraph 2.1. is not a standard section for articles in MDPI journals. I suggest that authors do not devote a whole paragraph to the list of equipment, but indicate the equipment as it is used in the relevant methods. A similar recommendation for paragraph 2.2.
- It is necessary to indicate the physicochemical characteristics of the soil used in the potting experiment.
- It is desirable to create a paragraph "Objects" in the methodological part of the article, where you can describe the sources of obtaining and a brief description of the bacteria and plants used in the work.
- In paragraph 2.5.1, you must provide a reference to the methodology.
- The titer of bacteria for inoculation is not indicated in the methodological part of the work.
- The authors write that they used the statistical method ANOVA. In this regard, the question arises: "Did the authors check the normality of the sample distribution and by what methods and using what criteria?" There is no information about this in the article.
- Lines 199-201 from the template. They need to be removed.
- The authors need to work on the dimensionality of numbers in the article. For example, 80.058% is unacceptable, it needs to be rounded.
- I would like the authors to correlate the data on the phosphorus content in plant tissues and in the rhizosphere with the standards for the maximum permissible concentration of phosphorus in soils and plant tissues.
- The list of references must be formatted according to the rules of the journal.
Only after revision can the article be published in the journal "Biology". Respectfully Yours, reviewer.
August 21, 2025
Author Response
- Key words should not duplicate words included in the title of the article.
Response: Thank you for your suggestion. As suggested by reviewer, we have re-chosen the key words in the revised manuscript (Line 53-54).
- The Introduction does not contain enough information about the role of the studied saponins in the life of the plant. I consider it necessary to add this information to the article.
Response: Thank you for your suggestion. As suggested by reviewer, we have added the suggested content to the manuscript (Line 95-99).
- It is desirable to more clearly define the purpose of the study in the Introduction. In its current form, it is written rather vaguely in lines 94-102 and after reading this text it is difficult to understand what was most important in this study.
Response: We are so sorry for our negligence, we have re-written this part according to your advise (Line 108-117).
- Paragraph 2.1. is not a standard section for articles in MDPI journals. I suggest that authors do not devote a whole paragraph to the list of equipment, but indicate the equipment as it is used in the relevant methods. A similar recommendation for paragraph 2.2.
Response: We think this is an excellent suggestion, we have made revision in the manuscript according to your advise to indicate the equipment as it is used in the relevant methods.
- It is necessary to indicate the physicochemical characteristics of the soil used in the potting experiment.
Response: Thank you for your careful review. We are so sorry for our negligence, and we have added these data to the article according to the reviewer’s suggestion (Line 127-129).
- It is desirable to create a paragraph "Objects" in the methodological part of the article, where you can describe the sources of obtaining and a brief description of the bacteria and plants used in the work.
Response: Thank you for you advice, we have made adjustments to the section “Experimental Design”, and the revised section is “2.1 Object and Experimental Design”, the relevant content in the section has also been adjusted accordingly (Line 119-144).
- In paragraph 2.5.1, you must provide a reference to the methodology.
Response: We are so sorry for our negligence, we have checked it carefully and provided the reference to the methodology (Line 164, Line 543-544).
- The titer of bacteria for inoculation is not indicated in the methodological part of the work.
Response: We are extremely grateful to reviewer for pointing out this problem. We have made a adjustment to the section “Experimental Design”, the concentration of bacterial suspension of each strain was 106 CFU·mL-1 (Line 137-138).
- The authors write that they used the statistical method ANOVA. In this regard, the question arises: "Did the authors check the normality of the sample distribution and by what methods and using what criteria?" There is no information about this in the article.
Response: Thank you for your careful review. We used the method of Shapiro-Wilk to check the normality of the sample distribution, if the P-value is greater than 0.05, it is considered that the data follows a normal distribution (Line 206-210).
- Lines 199-201 from the template. They need to be removed.
Response: We apologize for our negligence, we have replaced these sentence according to the reviewer’s suggestion.
- The authors need to work on the dimensionality of numbers in the article. For example, 80.058% is unacceptable, it needs to be rounded.
Response: Thank you for you suggestion, we have checked the numbers in the article, and in the data analysis of the article, certain numbers are precise values calculated by software, and if they are rounded, it will affect the subsequent data processing and thus affect the results.
- I would like the authors to correlate the data on the phosphorus content in plant tissues and in the rhizosphere with the standards for the maximum permissible concentration of phosphorus in soils and plant tissues.
Response: Thank you for your suggestion. As suggested by reviewer, we conducted a correlation analysis on the data of plant and soil phosphorus contents according to the reviewer’s advise, the result showed the content of phosphorus in plant was extremely significantly positively correlated with the contents of Ca8-P (P < 0.01), inorganic phosphorus and total hosphorus in soil and was significantly negatively correlated with the O-P content in soil (P < 0.05) (Line 355-376).
- The list of references must be formatted according to the rules of the journal.
Response: Thank you for you suggestion, we have made revision on the format of references according to the reviewer’s advice.

Reviewer 3 Report
Comments and Suggestions for Authors
The manuscript explains “Effects of inorganic phosphorus solubilizing bacteria on the rhizosphere phosphorus form and steroid saponin content of Paris polyphylla var. yunnanensis”. After a careful review, I found this work is not worthy to publish in the Biology in its current form. I propose this work can reconsider for publication after major revision as follow:
- Line 19: In abstract, the authors should replace word “To promote the phosphorus…” to “To improve the phosphorus..”
- Line 59: The authors should replace word “other nutrient” to “other nutrients”.
- Line 59: The authors should replace word “Pseudomonasand” to “Pseudomonas”.
- Line 71: The authors should add modern family “Melanthiaceae” along with family Liliaceae
- In materials and methods, the authors should rewrite the “reagent sections”.
- In material and methods, the authors should present the pre-sowing analysis of soil for physio-chemical and biological properties.
- In materials and methods, the authors should provide the references for standard reference solutions, sample solutions and chromatographic conditions.
- What is the reason to prepare different assay for S6?
- Line 185: In materials and methods, the authors should correct the word “above nethod” to “above method”
- In materials and methods, why did the authors not present other chromatographs of all applied treatments?
- Line 199-201: In results, the authors used some software for results interpretation “This section may be… can be drawn”. The authors should remove these sentences from the manuscript.
- The authors should rewrite the results section to be more interpretative and concise.
- The authors should tighten the discussion section by removing repetitive statements, concise and focusing on the most compelling interpretations of your data, especially the link between aryabhattai, soil P dynamics, and saponin synthesis discussion.
- In conclusion, the authors should clearly mention that the inorganic phosphorus solubilizing bacteria increase phosphorus in soil and uptake by plants. Why are the authors using word “could” again and again in their results.
- The authors should reduce the plagiarism from 31% to less than 15% in whole manuscript.
- There are many English grammar and sentence errors in whole manuscript. Kindly check it and correct it.
Author Response
- Line 19: In abstract, the authors should replace word “To promote the phosphorus…” to “To improve the phosphorus..”
Response: Thank you very much for your advise, we have revised that part according to your suggestion (Line 35).
- Line 59: The authors should replace word “other nutrient” to “other nutrients”.
Response: Thank you for your careful review. We have replaced “other nutrient” with “other insoluble elements” in the revised manuscript (Line 73).
- Line 59: The authors should replace word “Pseudomonasand” to “Pseudomonas”.
Response: Thank you for your careful review. We apologize for our negligence, we have replaced the word “Pseudomonasand” with “Pseudomonas” at that place (Line 82).
- Line 71: The authors should add modern family “Melanthiaceae” along with family Liliaceae
Response: We deeply appreciate the reviewer’s suggestion. According to the reviewer’s comment, we have added the modern family “Melanthiaceae” along with family Liliaceae in the revised manuscript (Line 84).
- In materials and methods, the authors should rewrite the “reagent sections”.
Response: We are extremely grateful to reviewer for pointing out this problem. We have re-written this part according to you suggestion, in the revised manuscript, we did not devote a whole paragraph to the list of equipment and reagent, but indicate the equipment as it is used in the relevant methods.
- In material and methods, the authors should present the pre-sowing analysis of soil for physio-chemical and biological properties.
Response: Thank you for your careful review. We are so sorry for our negligence, and we have added these data to the article according to the reviewer’s suggestion (Line 127-129).
- In materials and methods, the authors should provide the references for standard reference solutions, sample solutions and chromatographic conditions.
Response: Thank you for your advise, we have added the reference information to the corresponding position in the revised manuscript (Line 164, Line 173, Line 184, Line 543-544).
- What is the reason to prepare different assay for S6?
Response: Thank you for your careful review. In this study, we randomly selected the treatment group S6 for methodological validation.
- Line 185: In materials and methods, the authors should correct the word “above nethod” to “above method”
Response: We are so sorry for our negligence, we have corrected that word to “method described” in the manuscript (Line 196).
- In materials and methods, why did the authors not present other chromatographs of all applied treatments?
Response: We appreciate the reviewer’s careful review for our work, in this manuscript, we only provided the chromatograph of standard reference and sample S6 in the section Materials and Methods to show the shape and positional relation of the different saponin peaks, more importantly, in this section, the information we provided can show that the method we adopted is effective. However, providing chromatograms of all samples in the results would be too verbose.
- Line 199-201: In results, the authors used some software for results interpretation “This section may be… can be drawn”. The authors should remove these sentences from the manuscript.
Response: We apologize for our negligence, we have replaced these sentence according to the reviewer’s suggestion.
- The authors should rewrite the results section to be more interpretative and concise.
Response: We are grateful for the suggestion. We have re-written the results section according to the reviewer’s advise.
- The authors should tighten the discussion section by removing repetitive statements, concise and focusing on the most compelling interpretations of your data, especially the link between aryabhattai, soil P dynamics, and saponin synthesis discussion.
Response: Thank you for your suggestion, we have made revision according to the reviewer’s suggestion (Line 408-463).
- In conclusion, the authors should clearly mention that the inorganic phosphorus solubilizing bacteria increase phosphorus in soil and uptake by plants. Why are the authors using word “could” again and again in their results.
Response: Thank you for you adivise. We agree with the comment and re-wrote the conclusion section (Line 464-485).
- The authors should reduce the plagiarism from 31% to less than 15% in whole manuscript.
Response: Thank you for your careful review, we have made revision and reduced the overall similarity to 17%, and the duplicate check report is provided in the attachment.
- There are many English grammar and sentence errors in whole manuscript. Kindly check it and correct it.
Response: Thank you for your careful review. We are very sorry for the mistakes in this manuscript and inconvenience they caused in your reading. The manuscript has been thoroughly revised and edited by a professional company, so we hope it can meet the journal’s standard. In addition, we provided the editing certificate in the attachment.

Round 2
Reviewer 3 Report
Comments and Suggestions for Authors
The authors have addressed most of my previous comments and suggestions. Prior to publication, the following minor revisions are required:
All chromatograms for the steroid saponin analysis must be included in the supplementary data to ensure full transparency and reproducibility.
The resolution and legibility of several figures require improvement to meet the journal's publication standards.
I recommend acceptance once these revisions have been completed.
Author Response
- All chromatograms for the steroid saponin analysis must be included in the supplementary data to ensure full transparency and reproducibility.
Response: Thank you very much for your advise, we have provided all the chromatograms for the steroid saponin analysis in the supplementary (Line 188, Supplementary File, Figure S1 - S8).
- The resolution and legibility of several figures require improvement to meet the journal's publication standards.
Response: Thank you for your careful review. We have improved the resolution and legibility of Figure 1 in the manuscript (Figure 1).
